# MOTIONGRPO: Overcoming Low Intra-Group Diversity in GRPO-Based Egocentric Motion Recovery

**Nanjie Yao** [1]   **Junlong Ren** [1]   **Wenhao Shen** [2]   **Hao Wang** [1]

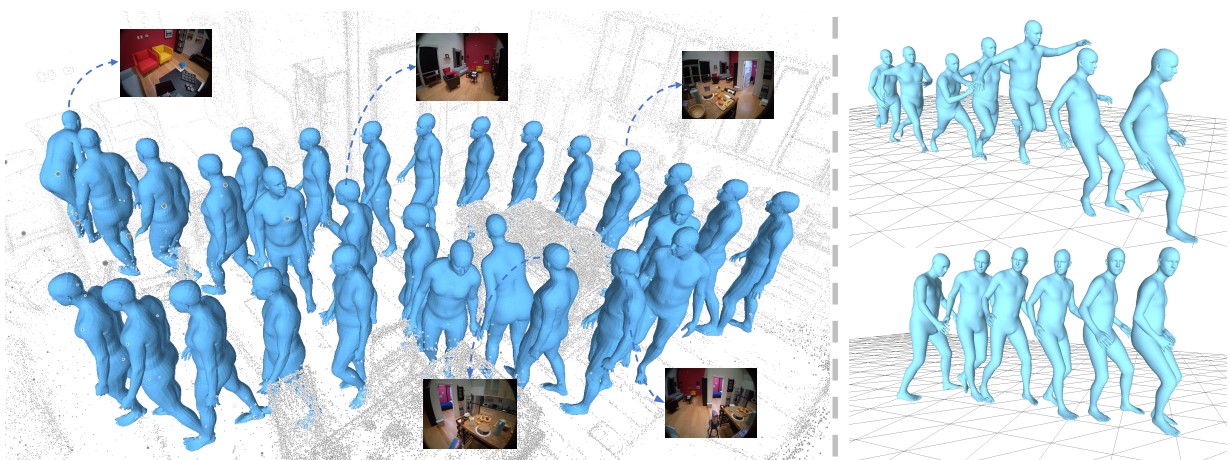

*Figure 1.* We introduce MOTIONGRPO, a RL-based framework designed to provide fine-grained geometric and visual guidance. Given the head trajectory signals and optional egocentric images, our method recovers high-fidelity full-body human motion in 3D scenes.

## Abstract

This paper studies full-body 3D human motion recovery from head-mounted device signals. Existing diffusion-based methods often rely on global distribution matching, leading to local joint reconstruction errors. We propose MOTIONGRPO, a novel framework leveraging reinforcement learning post-training to inject fine-grained guidance into the diffusion process. Technically, we model diffusion sampling as a Markov decision process optimized via Group Relative Policy Optimization (GRPO). To this end, we introduce a hybrid reward mechanism that combines a learned conditioned perceptual model for global visual plausibility and explicit constraints for local joint precision. Our key technical insight is that policy optimization in diffusion-based recovery suffers from vanishing gradients due to limited intra-group sample diversity. To address this, we further introduce a noise-injection strategy that explicitly increases sample variance and stabilizes learning. Extensive experiments demonstrate that MOTIONGRPO achieves state-of-the-art performance with superior visual fidelity. Code is available at: `https://github.com/3DAgentWorld/MotionGRPO/`

## 1. Introduction

Recovering full-body 3D human motions from head trajectory signals captured by Head-Mounted Devices (HMDs) such as Project Aria (Engel et al., 2023) remains a critical challenge for applications in VR/AR. Unlike third-person motion capture (Shen et al., 2025a; 2024), egocentric settings suffer from severe occlusion, where the user's body is largely unobserved by the front-facing cameras. Consequently, estimating accurate full-body pose requires strong priors to resolve kinematic ambiguities and ground the motion in the physical world.

Existing approaches (Li et al., 2023; Yi et al., 2025) adopt diffusion-based generative frameworks (Ho et al., 2020; Song et al., 2021) to model distributions of plausible human motions. During the training phase, ground-truth motion

[1]The Hong Kong University of Science and Technology (Guangzhou), China [2]Nanyang Technological University, Singapore. Correspondence to: Hao Wang <haowang@hkust-gz.edu.cn>.

*Proceedings of the 43rd International Conference on Machine Learning*, Seoul, South Korea. PMLR 306, 2026. Copyright 2026 by the author(s).

sequences are corrupted with Gaussian noise and a conditional denoising network learns a motion prior conditioned on head trajectories. At inference, full-body motions are recovered via an iterative reverse diffusion process starting from Gaussian noise.

Despite strong generative ability, diffusion-based methods struggle with fine-grained joint control and visual fidelity. They often produce spatial misalignment between predicted and ground-truth joints. Moreover, visual artifacts such as foot skating, motion jitter, and ground penetration are also common. These issues arise from the difficulty of enforcing explicit geometric constraints in diffusion models. During early denoising steps, poses are heavily corrupted and unrealistic. As a result, direct joint-level supervision becomes unstable or ineffective. Diffusion objectives therefore focus on distribution matching rather than precise joint alignment. Improving joint position accuracy and visual fidelity under this framework remains challenging.

To address these limitations, we propose MOTIONGRPO, a Reinforcement Learning (RL) post-training framework for diffusion-based motion recovery. It injects fine-grained guidance into the diffusion sampling process and improves visual plausibility. We formulate diffusion sampling as a Markov Decision Process (MDP) and optimize it via Group Relative Policy Optimization (GRPO) within a Stochastic Differential Equation (SDE)-based diffusion framework. A hybrid reward is designed to adapt this generation-oriented optimization scheme to reconstruction tasks.

The hybrid reward focuses on both global visual plausibility and local joint accuracy. For global guidance, we introduce a trajectory-conditioned perceptual model to evaluate visual plausibility. The model is trained with online contrastive learning and actively synthesizes hard negative samples. This enables reliable detection of visual artifacts such as foot skating and motion jitter that are often missed by standard losses. For local precision, we incorporate explicit sub-rewards on joint positions, rotations, and velocities. Together, these rewards guide the diffusion model toward both visually plausible motions and accurate joint alignment.

Our key technical insight is that directly applying GRPO to diffusion-based motion recovery suffers from vanishing gradients. In standard generative tasks (Rombach et al., 2022; Tevet et al., 2023; Achiam et al., 2023; Liu et al., 2024), models generate diverse outputs, providing sufficient variance for advantage normalization. In contrast, motion recovery is strongly conditioned on head signals, which constrains the output space and reduces intra-group sample diversity. To address this bottleneck, we introduce a noise-injection strategy. Temporally smoothed noise is added to the input conditions to simulate out-of-distribution inputs. This increases model uncertainty and output diversity, which is crucial for effective GRPO optimization.

Our key contributions can be summarized as follows:

- We propose MOTIONGRPO, an RL-based framework for egocentric motion recovery. It optimizes a hybrid reward combining a trajectory-conditioned perceptual model for global plausibility and fine-grained objectives for precise joint alignment.

- We identify the "low intra-group diversity" bottleneck in GRPO for motion recovery tasks. To mitigate vanishing gradients, we introduce a temporally smoothed noise-injection strategy that increases output diversity and stabilizes training.

- Extensive experiments on AMASS and RICH benchmarks show that MOTIONGRPO achieves state-of-the-art performance. Qualitative results further demonstrate strong generalization and real-world scalability.

**Conflict of Interest Disclosure.** The authors declare that they have no conflicts of interest to disclose.

## 2. Related Works

**Egocentric Human Motion Recovery.** Egocentric human motion recovery aims to reconstruct full-body motion from sparse observations. Existing approaches generally fall into two categories: multi-sensor setups and HMD-only methods. The former utilizes distributed Inertial Measurement Units (Kim & Lee, 2022; Yi et al., 2021; Lee & Joo, 2024; Yi et al., 2023) or hand controllers (Castillo et al., 2023; Jiang et al., 2022; Du et al., 2023) to achieve high accuracy but is limited by the intrusive hardware setup. The latter focuses on recovering global motion solely from HMD signals, offering better practicality. Early works in this category (Yuan & Kitani, 2019; Luo et al., 2021) employed regression-based networks (Hochreiter & Schmidhuber, 1997). Recent state-of-the-art methods have shifted toward generative models. Notably, EgoEgo (Li et al., 2023) decouples the task into SLAM-based head pose estimation and diffusion-based body synthesis, enabling the use of large-scale unpaired datasets. Based on this, EgoAllo (Yi et al., 2025) introduces invariant conditioning and visual hand observations, significantly enhancing generalization and scene-relative stability. MOTIONGRPO extends EgoAllo by collaborating generative models with RL-based post-training, specifically aiming to resolve the fine-grained control challenges in diffusion-based motion recovery methods.

**Reinforcement Learning in 3D Human.** Reinforcement Learning (RL) has become pivotal in aligning generative models (Wallace et al., 2024; Black et al., 2024; Guo et al., 2025) with human preferences through methods such as Reinforcement Learning from Human Feedback (Christiano

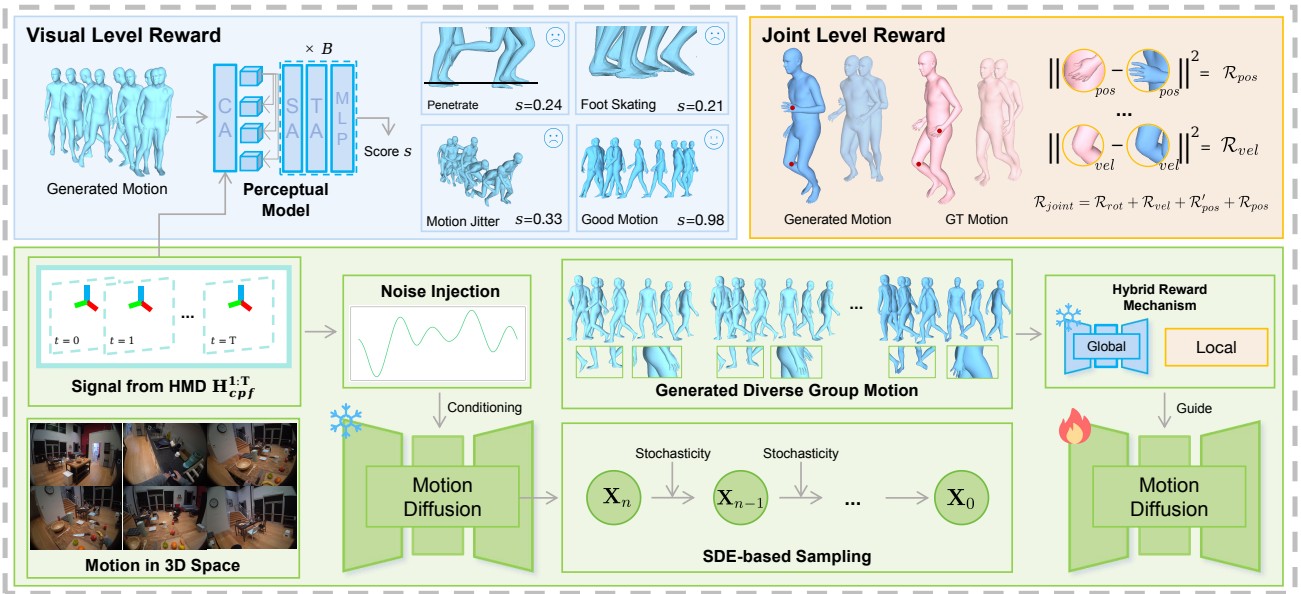

Figure 2. **Method Overview.** Given the input head trajectory signals $\mathbf{H}_{cpf}^{1:T}$ from HMD, we employ them as conditions for a motion diffusion model to recover human motion. To address low intra-group diversity, we obtain diverse motion outputs through SDE-based sampling and noise injection on trajectory conditions. Based on these outputs, we utilize a proposed hybrid reward mechanism for comprehensive reward calculation: at the global visual level, a trajectory-conditioned perceptual model via spatial-temporal attention assesses visual plausibility; at the local joint level, we enforce explicit joint constraints to align generated results with GT. Finally, we calculate advantages and use GRPO to update the model parameters, providing guidance for both visual plausibility and joint accuracy.

et al., 2017) and Direct Preference Optimization (Rafailov et al., 2023). In human-centric domains, RL is commonly applied in tasks like text-guided motion generation (Yuan et al., 2023; Han et al., 2025) and monocular human pose estimation (Shen et al., 2025a) to generate motions that ensure physical validity or align with human preferences. Regarding the specific task of egocentric motion recovery, a small fraction of existing works (Yuan & Kitani, 2019) have utilized traditional RL-based techniques such as Proximal Policy Optimization (Schulman et al., 2017) to recover human motion within physics simulators (Todorov et al., 2012). However, these RL methods often suffer from training instability or high training cost. Recently, GRPO (Shao et al., 2024) has emerged as a superior alternative, eliminating the need for a separate value network by estimating baselines directly from group-wise scores. Nevertheless, unlike generation tasks that benefit from high variance, reconstruction is heavily constrained by inputs, leading to low intra-group diversity that hinders vanilla GRPO training. MOTIONGRPO bridges this gap by adapting GRPO with a novel noise-injection strategy to enable robust post-training for diffusion-based motion recovery.

## 3. Methodology

### 3.1. Preliminary

**Diffusion Model for Motion Recovery.** The forward process of a diffusion model (Ho et al., 2020) is defined as a fixed Markov chain that gradually adds Gaussian noise to the data $\mathbf{x}_0$ according to a variance schedule $\sigma_t \in (0, 1)$. At any timestep $t$, the noisy latent $\mathbf{x}_t$ can be sampled directly via $\mathbf{x}_t = \sqrt{\bar{\alpha}_t}\mathbf{x}_0 + \sqrt{1 - \bar{\alpha}_t}\boldsymbol{\epsilon}$, where $\bar{\alpha}_t = \prod_{s=1}^{t}(1 - \sigma_s)$ and $\boldsymbol{\epsilon} \sim \mathcal{N}(\mathbf{0}, \mathbf{I})$. Generative sampling aims to reverse this forward process, which can be expressed as:

$$p_\theta(\mathbf{x}_{t-1}|\mathbf{x}_t) = \mathcal{N}(\mathbf{x}_{t-1}; \boldsymbol{\mu}_\theta(\mathbf{x}_t, t), \boldsymbol{\Sigma}_\theta(\mathbf{x}_t, t)), \quad (1)$$

where $p_\theta$ represents the learned reverse transition distribution. Adopting the previous framework (Yi et al., 2025), we model this sampling process with an ego-condition as:

$$p_\theta(\mathbf{x}_{t-1}|\mathbf{x}_t, \mathbf{c}) = \mathcal{N}(\mathbf{x}_{t-1}; \boldsymbol{\mu}_\theta(\mathbf{x}_t, t, \mathbf{c}), \sigma_t^2\mathbf{I}). \quad (2)$$

Here, $\mathbf{c}$ denotes the condition of the head trajectory (Detailed in Sec. 3.2). This method trains a transformer $\boldsymbol{\mu}_\theta$ to predict the original sample from the noisy sample $\mathbf{x}_t$ and the condition $\mathbf{c}$. Typically, this network is optimized to minimize the weighted squared error between the predicted original sample and the Ground Truth (GT) $\mathbf{x}_0$. The diffusion training loss is formulated as:

$$\mathcal{L} = \min_\theta \mathbb{E}_{\mathbf{x}_0, t}\left[w_t\|\boldsymbol{\mu}_\theta(\mathbf{x}_t, t, \mathbf{c}) - \mathbf{x}_0\|^2\right], \quad (3)$$

where $t$ is sampled uniformly from the diffusion timesteps and $w_t$ is a noise-dependent weighting term.

**Group Relative Policy Optimization.** Group Relative Policy Optimization (GRPO) (Shao et al., 2024) extends the

Proximal Policy Optimization (Schulman et al., 2017) framework to a group-wise formulation. GRPO estimates the learned value baseline from the group scores of multiple sampled outputs. Specifically, for each query $q$, a group of outputs $\{o_i\}_{i=1}^G$ is sampled from the old policy $\pi_{\theta_{old}}$. GRPO optimizes the policy model $\pi_\theta$ by maximizing the following objective (we omit the clip term and Kullback-Leibler term hereinafter for brevity):

$$\mathcal{J}_{GRPO}(\theta) = \mathbb{E}_{q,\{o_i\}_{i=1}^G \sim \pi_{old}(\cdot|q)} \left[ \frac{1}{G} \sum_{i=1}^G \left( \frac{\pi_\theta(o_i|q)}{\pi_{old}(o_i|q)} \hat{A}_i \right) \right],$$
(4)

where the advantage term $\hat{A}_i$ is computed using the group-relative normalization of rewards $\{\mathcal{R}_i\}_{i=1}^G$ corresponding to the samples within the group:

$$\hat{A}_i = \frac{\mathcal{R}_i - \text{mean}(\{\mathcal{R}_1, \ldots, \mathcal{R}_G\})}{\text{std}(\{\mathcal{R}_1, \ldots, \mathcal{R}_G\})}.$$
(5)

### 3.2. Problem Formulation

Given a sequence of $T$ timesteps, our goal is to recover the human's full-body 3D motion. Let $\mathbf{H}_{cpf}^{1:T} = \{R_{cpf}^{1:T}, \tau_{cpf}^{1:T}\} \in$ SE(3) denotes the millimeter-level accurate raw signals of *central pupil frames* (CPF) captured by SLAM systems of devices like Project Aria (Pan et al., 2023), where $R_{cpf}^{1:T}$ and $\tau_{cpf}^{1:T}$ represent the rotation and global translation, respectively. Our model, MOTIONGRPO, $\mathcal{F}(\cdot)$ takes $\mathbf{H}_{cpf}$ as input and produces the reconstructed full-body human motion $\mathbf{M}$ as output:

$$\mathbf{M}^{1:T} = \mathcal{F}(\mathbf{c}^{1:T}), \text{ where } \mathbf{c}^{1:T} = g(\mathbf{H}_{cpf}^{1:T}).$$
(6)

Here, $\mathbf{M}^{1:T} = \{\Theta^{1:T}, \beta^{1:T}\}$ follows the standard SMPL-H representation (Romero et al., 2017), where $\Theta \in \mathbb{R}^{51 \times 3 \times 3}$ denotes the local joint rotations and $\beta \in \mathbb{R}^{16}$ denotes the temporal invariant body shape parameters. The $g(\cdot)$ is the invariant conditioning function (Yi et al., 2025). These parameters are subsequently processed with $\mathbf{H}_{cpf}^{1:T}$ via forward kinematics to generate global human motions. We implement the model $\mathcal{F}$ via a transformer-based diffusion architecture (Vaswani et al., 2017). This design allows the model to denoise the motion $\mathbf{M}$ from Gaussian noise conditioned on the head trajectory progressively. The overview of the proposed MOTIONGRPO can be seen in Fig. 2.

### 3.3. Vanilla GRPO for Motion Recovery

Following previous works (Black et al., 2024), we model the sampling process of the motion diffusion model as a multi-step MDP defined by the tuple $(\mathcal{S}, \mathcal{A}, \pi, \mathbf{R})$, which represents the state space, action space, policy and reward functions, respectively. The state at sampling timestep $t \in \{n, n-1, \ldots, 0\}$ is defined as $s_t = \{(\mathbf{c}, t, \mathbf{x}_t)|s_t \in \mathcal{S}\}$, comprising the head condition, current timestep, and noisy

motion latent. The action is defined as the denoised sample at the next step, $a_t = \{\mathbf{x}_{t-1}|a_t \in \mathcal{A}\}$. Consequently, the policy $\pi_\theta(a_t|s_t)$ corresponds to the parameterized reverse transition distribution $p_\theta(\mathbf{x}_{t-1}|\mathbf{x}_t, \mathbf{c})$. The reward is sparse and assigned only at the final timestep $t = 0$, such that $\mathcal{R}(s_t, a_t) = \mathbf{R}(\mathbf{x}_0, \mathbf{c})$ if $t = 0$.

To enable GRPO, following (Xue et al., 2025), we utilize SDE-based sampling to introduce necessary stochasticity to generate a diverse group of outputs $\{o_i\}_{i=1}^G$ for advantage estimation while ensuring training stability by assigning shared initialization noise across the group. Specifically, let $\mathbf{f}(\mathbf{x}, t)$ and $\varphi(t)$ denote the drift and diffusion coefficients of the forward process. The forward SDE can be represented by $d\mathbf{x} = \mathbf{f}(\mathbf{x}, t)dt + \varphi(t)d\mathbf{w}$. The corresponding reverse SDE can be expressed as:

$$d\mathbf{x}_t = (\mathbf{f}(\mathbf{x}, t) - \frac{1 + \epsilon_t^2}{2}\varphi(t)^2 \nabla_{\mathbf{x}_t} \log p_t(\mathbf{x}_t))dt + \epsilon_t d\mathbf{w},$$
(7)

where $d\mathbf{w}$ is the standard Wiener process, and $\epsilon_t$ is the introduced stochasticity at sampling timestep $t$. For each $o$, we compute the rewards with $\{\mathbf{R}_k\}_{k=1}^K$.

Following Eq. 5, we compute the corresponding advantage $\hat{A}_{i,k}$ for each reward function independently. The aggregated advantage is defined as the summation of these component advantages, denoted as $\hat{A}_i = \sum_{k=1}^K \hat{A}_{i,k}$. The final optimization objective aggregates gradients across the sampled timesteps and group members:

$$\mathcal{J}_{\text{GRPO}}(\theta) = \mathbb{E}_{\mathbf{c}, \{o_i\} \sim \pi_{\text{old}}(\cdot|\mathbf{c})} \left[ \frac{1}{G} \sum_{i=1}^G \frac{1}{n} \sum_{t=1}^n \left( \frac{\pi_\theta(o_{i,t}|\mathbf{c})}{\pi_{\text{old}}(o_{i,t}|\mathbf{c})} \hat{A}_i \right) \right].$$
(8)

### 3.4. Hybrid Reward Mechanism

#### 3.4.1. VISUAL LEVEL REWARD[1]

Given a group of generated human motion sequences and their corresponding SLAM-derived head trajectories, denoted as $\{o_i\}_{i=1}^G = \{(\mathbf{M}^{1:T}, \mathbf{H}_{cpf}^{1:T})_i\}_{i=1}^G$, the objective of the perceptual model is to evaluate plausibility scores for all pairs, denoted as $\{s_i\}_{i=1}^G$. A higher score indicates a motion sequence that is not only naturally plausible but also geometrically consistent with the ego-centric head trajectory.

**Model Architecture.** We firstly process the input motion sequences into an SMPL-H skeleton representation $\mathbf{J} \in \mathbb{R}^{T \times N \times D}$ via forward kinematics, denoted as $\mathbf{J} \in \mathbb{R}^{T \times N \times D}$, where $N = 21$ is the number of body joints and $D = 7$ represents the quaternion representation of the rotation at each joint along with the global position. These skeletons and the corresponding head trajectories are projected

---

[1] We term this reward "visual" because its primary objective is to evaluate visually perceivable plausibility and mitigate unnatural visual artifacts.

into a latent feature space using frame-wise and keypoint-wise linear embedding layers, obtaining the corresponding latent features $\mathbf{F}_\mathrm{J} \in \mathbb{R}^{T \times N \times d}$ and $\mathbf{F}_\mathrm{H} \in \mathbb{R}^{T \times N \times d}$, where $d$ is the latent dimension. Subsequently, we introduce a Cross-Attention (CA) mechanism to fuse these heterogeneous modalities. The fused features are then processed by a Transformer-based encoder with $B$ blocks, where each block alternates between MLP, Spatial-Attention (SA), and Temporal-Attention (TA) layers (Zhang et al., 2024). Finally, these features are decoded into scores by an MLP-based network and normalized using the sigmoid function.

**Contrastive Training.** To achieve the goal of perceptual scoring, we adopt an online contrastive learning approach. Unlike the offline method by adding unreal noise to the GT samples (Shen et al., 2025a), we actively synthesize hard negative samples on-the-fly to challenge the perceptual model. Concretely, given a batch of head trajectories, we utilize the base policy model to generate a set of skeleton sequences. The pairs of sequence and the corresponding head trajectory serve as negative samples. To prevent over-fitting to a single deterministic output and to enrich the diversity of the negative set, we randomly extract outputs from the last three sampling timesteps. Subsequently, we treat the pair of the GT motion sequence and its head tracking data as positive samples. We concatenate them with the generated negative samples and feed the combined batch into the model. The model is then optimized using the In-foNCE (Oord et al., 2018) loss. Formally, the loss term is defined as follows:

$$\mathcal{L}_{\mathrm{NCE}} = -\mathbb{E}\left[\log \frac{\exp(\phi(\mathbf{J}^+|\mathbf{H}^+)/\delta)}{\exp(\phi(\mathbf{J}^+|\mathbf{H}^+)/\delta) + \sum_{i=1}^{\mathbf{N}} \exp(\phi(\mathbf{J}_i^-|\mathbf{H}_i^-)/\delta)}\right],$$
(9)

where $\mathbf{H}$ represents the head trajectory condition, $\mathbf{J}^+$ denotes the positive GT sample, and $\{\mathbf{J}_i^-\}_{i=1}^n$ represents the set of $n$ generated negative samples, and $\delta = 0.07$ is the temperature hyperparameter scaling the distribution.

**Reward Formulation.** Once trained, the perceptual model $\phi(\cdot)$ provides the feedback signal for the reinforcement learning stage. The final visual level reward $\mathcal{R}_{vis}$ is derived from the predicted score $s$ via:

$$\mathcal{R}_{vis} = \exp(\omega_{vis} \cdot s),$$
(10)

where $\omega_{vis}$ is the weighting coefficient.

### 3.4.2. JOINT LEVEL REWARD

In addition to the learnable visual reward, we design a composite metric-based reward function (as detailed in Appendix D) to ensure high fidelity to the GT motion, leveraging the GRPO algorithm's ability to optimize non-differentiable objectives. We introduce four sub-rewards:

$$\mathcal{R}_{rot} = \exp\left[-\frac{\omega_{rot}}{T} \sum_{\mathcal{T}=1}^{T} \left(\frac{1}{N} \sum_{j=1}^{N} \|\mathbf{r}_{\mathcal{T},j} - \hat{\mathbf{r}}_{\mathcal{T},j}\|_1\right)\right],$$
(11)

$$\mathcal{R}_{pos} = \exp\left[-\frac{\omega_{pos}}{T} \sum_{\mathcal{T}=1}^{T} \left(\frac{1}{N} \sum_{j=1}^{N} \|\mathbf{p}_{\mathcal{T},j} - \hat{\mathbf{p}}_{\mathcal{T},j}\|_2\right)\right],$$
(12)

$$\mathcal{R}'_{pos} = \exp\left[-\frac{\omega'_{pos}}{T} \sum_{\mathcal{T}=1}^{T} \left(\frac{1}{N} \sum_{j=1}^{N} \|\mathbf{p}'_{\mathcal{T},j} - \hat{\mathbf{p}}'_{\mathcal{T},j}\|_2\right)\right],$$
(13)

$$\mathcal{R}_{vel} = \exp\left[-\frac{\omega_{vel}}{T} \sum_{\mathcal{T}=1}^{T} \left(\frac{1}{N} \sum_{j=1}^{N} \|\mathbf{v}_{\mathcal{T},j} - \hat{\mathbf{v}}_{\mathcal{T},j}\|_2\right)\right],$$
(14)

where $\omega_{rot}$, $\omega_{pos}$, $\omega'_{pos}$, and $\omega_{vel}$ are weight coefficients, and $\hat{\cdot}$ denotes the GT motion.

Specifically, $\mathcal{R}_{pos}$ and $\mathcal{R}'_{pos}$ represent pose rewards computed with and without per-frame similarity transform alignment (Umeyama, 2002), respectively. $\mathcal{R}_{rot}$ quantifies local rotation discrepancies, and $\mathcal{R}_{vel}$ penalizes velocity differences. Here, $\mathbf{p}_{\mathcal{T},j}$ and $\mathbf{p}'_{\mathcal{T},j}$ denote the global position of the $j$-th joint at frame $\mathcal{T}$ (before and after alignment), $\mathbf{r}_{\mathcal{T},j}$ is the local rotation, and $\mathbf{v}_{\mathcal{T},j}$ is the velocity.

The total reward is summarized as:

$$\mathcal{R}_{total} = \mathcal{R}_{vis} + \mathcal{R}_{joint},$$
(15)

where $\mathcal{R}_{joint} = \mathcal{R}_{rot} + \mathcal{R}_{pos} + \mathcal{R}'_{pos} + \mathcal{R}_{vel}$.

### 3.5. Mitigating Low Intra-Group Diversity

#### 3.5.1. LOW INTRA-GROUP DIVERSITY

With GRPO, we can improve model performance to some extent. However, directly applying this paradigm to motion recovery presents a fundamental challenge due to the deterministic nature of the task. Unlike open-ended generation tasks where the policy is encouraged to explore diverse modes, egocentric motion recovery is heavily constrained by the strong conditioning of the head trajectory inputs $\mathbf{c}$. Concretely, the sampled outputs $\{o_i\}_{i=1}^{G}$ within a single group tend to exhibit high similarity and minimal variance. This lack of diversity becomes critical when analyzing the advantage estimation mechanism in GRPO. Referring to Eq. 5, the advantage $\hat{A}_i$ relies on the group-relative normalization. When the intra-group diversity is low, the rewards $\{\mathcal{R}_i\}_{i=1}^{G}$ for the generated motions become nearly identical, causing the standard deviation term in the denominator to approach zero. This numerical instability makes the normalized advantages non-informative or explosive, leading to the vanishing gradient problem. As a result, the optimization objective in Eq. 8 fails to provide meaningful policy gradient updates, limiting the training process.

**Algorithm 1** MOTIONGRPO

**Require:** Human motion dataset $\mathcal{D}$; Denoiser Policy $\pi_\theta$; Reward Functions $\{\mathbf{R}_k\}_{k=1}^K$; Group Size $G$; Learning Rate $\eta$; Sampling Timesteps $n$.

**Ensure:** Optimize the denoiser policy.

1: **while** not converged **do**
2:     Sample batch of head trajectory and corresponding human skeletons $\mathcal{D}_b = \{\mathbf{H}_1, \ldots, \mathbf{H}_B\} \sim \mathcal{D}$
3:     Update policy denoiser: $\pi_{\theta_{old}} \leftarrow \pi_\theta$
4:     **for** each condition $\mathbf{H} \in \mathcal{D}_b$ **do**
5:         Perturb condition: $\mathbf{c} \leftarrow g(\tilde{\mathbf{H}})$.
6:         Generate $G$ samples $\{o_i\}_{i=1}^G$ using $\pi_\theta(\cdot|\mathbf{c})$ via sampling.
7:         **for** each reward function $k = 1, \ldots, K$ **do**
8:             Compute group statistics:
                $\mu_k = \text{mean}(\{\mathcal{R}_{i,k}\}_{i=1}^G), \sigma_k = \text{std}(\{\mathcal{R}_{i,k}\}_{i=1}^G)$
9:         **end for**
10:         **for** each sample $i = 1, \ldots, G$ **do**
11:             Compute advantage: $\hat{A}_{i,k} = \frac{1}{K}\sum_{k=1}^K \frac{\mathcal{R}_{i,k}-\mu_k}{\sigma_k}$
12:         **end for**
13:     **end for**
14:     **for** $t$ in range$(1, n)$ **do**
15:         Update policy: $\theta \leftarrow \theta + \eta\nabla_\theta\mathcal{J}_{GRPO-Motion}(\theta)$
16:     **end for**
17: **end while**

### 3.5.2. NOISE-INJECTION STRATEGY

To mitigate the vanishing gradient issue caused by low intra-group diversity, we propose a simple yet effective strategy: injecting temporally smoothed noise into the input conditions. The core insight is that the strong conditioning from accurate head signals overly constrains the policy, leading to a collapse in output diversity. By introducing controlled perturbations, we simulate pseudo out-of-distribution inputs. This forces the diffusion policy to face slightly shifted states, thereby increasing the model's predictive uncertainty and restoring the necessary variance among the group samples $\{o_i\}_{i=1}^G$. This artificially induced diversity ensures a non-trivial standard deviation in the advantage calculation, enhancing the effectiveness of RL optimization.

We specifically choose Perlin noise (Perlin, 2002) for this strategy to maintain the temporal smoothness inherent in physical head trajectories, avoiding high-frequency jitters that could disrupt the motion prior. We apply this noise specifically to the translation of the head condition. Formally, given the head condition $\mathbf{H} = \{R, \tau\}$ the perturbed condition $\tilde{\mathbf{H}}$ is formulated as:

$$\tilde{\mathbf{H}} = \{R, \tau + \lambda \cdot \mathcal{P}(t)\}, \tag{16}$$

where $\mathcal{P}(t)$ denotes the time-continuous Perlin noise sequence and $\lambda$ is a scaling factor controlling the noise magnitude. This perturbed input is then processed by the invariant

function $g(\cdot)$ to obtain the conditioning feature $\mathbf{c}$ for the diffusion process. The overview of GRPO training used in MOTIONGRPO can be seen in Algorithm 1.

## 4. Experiments

### 4.1. Experiment Setups

**Dataset.** For training data, our method requires sequences containing full-body motion with associated SMPL shape parameters for reward calculation, and head SLAM device poses as input conditions. Following prior work (Yi et al., 2025), we adopt the AMASS dataset (Mahmood et al., 2019) as our training data. For evaluation, we evaluate with two datasets: AMASS and RICH (Huang et al., 2022). Similar to training, we utilize the synthetic device pose for inference. To validate the model's performance in a real testbed, we additionally introduce the Aria Digital Twins (ADT) (Pan et al., 2023) for the visualization and qualitative evaluation.

**Metrics.** To comprehensively evaluate the quality of the recovered human motion, we employ a diverse set of metrics categorized into Joint Accuracy and Visual Quality and Plausibility. For Joint Accuracy, we report Mean Per-Joint Position Error (**MPJPE**) and Procrustes-Aligned MPJPE (**PA-MPJPE**) to evaluate 3D pose reconstruction error (in mm). We also report Mean Per-Joint Velocity Error (**MPJVE**) to assess temporal dynamics, and Mean Per-Joint Rotational Error (**MPJRE**) to measure joint rotation accuracy. Regarding Visual Quality and Plausibility, we employ **Jitter** to quantify motion smoothness (high-frequency noise), **Ground Penetration (GP)** to measure floor penetration, and **Foot Skating (FS)** to detect unnatural foot skating artifacts during ground contact. Additionally, we use the **Accuracy**, **Wrong Count** to evaluate the performance of the perceptual model and a variant of **Diversity** (Tevet et al., 2023) metric to evaluate the intra-group diversity. The detailed description of metrics can be seen in Appendix D.

**Baselines.** Direct comparison with many existing egocentric human motion recovery methods is often difficult due to the discrepancies in problem formulation and input modalities. Consequently, we establish baselines using EgoEgo (Li et al., 2023) and EgoAllo (Yi et al., 2025). To ensure experimental fairness, we standardize the input settings by restricting all methods to utilize only head trajectories.

### 4.2. Quantitative Evaluation

**Joint Accuracy.** Table 1 presents the quantitative comparisons with state-of-the-art egocentric motion recovery methods. MOTIONGRPO consistently establishes a new state-of-the-art across both AMASS and RICH benchmarks. On the AMASS dataset, our framework significantly outperforms the most competitive baseline, EgoAllo, reducing

*Table 1.* **Quantitative evaluation** on **AMASS** and **RICH** datasets. We evaluate Local Joint Accuracy using MPJPE, PA-MPJPE, MPJVE, and MPJRE. Global Visual Quality and Plausibility are measured by Jitter, Ground Penetration (GP), and Foot Skating (FS). "↓" indicates lower is better. The best and the second best results are highlighted with **bold** and underline. The superscript "$\aleph$" denotes that the results are post-processed with test-time (Yi et al., 2025). Note that all of the baselines are reproduced on the official implementations.

| Method | AMASS Dataset | | | | | | | RICH Dataset | | | | | | |
|---|---|---|---|---|---|---|---|---|---|---|---|---|---|---|
| | Joint Accuracy | | | | Visual Quality | | | Joint Accuracy | | | | Visual Quality | | |
| | MPJPE | PA-MPJPE | MPJVE | MPJRE | Jitter | GP | FS | MPJPE | PA-MPJPE | MPJVE | MPJRE | Jitter | GP | FS |
| | (mm)↓ | (mm)↓ | (mm)↓ | (°)↓ | ↓ | (m)↓ | (m)↓ | (mm)↓ | (mm)↓ | (mm)↓ | (°)↓ | ↓ | (m)↓ | (m)↓ |
| EgoEgo | 177.231 | 152.125 | 588.661 | 9.457 | 2.643 | 1.331 | 1.241 | 221.45 | 196.223 | 572.331 | 13.312 | 5.187 | 4.357 | 1.021 |
| EgoAllo | 124.985 | 103.958 | 553.221 | 8.777 | 2.394 | 1.143 | 1.290 | 192.686 | 172.724 | 506.992 | 12.734 | 4.135 | 4.145 | 1.094 |
| EgoAllo$^\aleph$ | 121.651 | 101.034 | 483.471 | 8.728 | 1.455 | 1.099 | 0.479 | 190.000 | 169.838 | 407.628 | 12.638 | 1.880 | 4.438 | 0.223 |
| MOTIONGRPO | 114.207 | 95.512 | 531.217 | 8.413 | 2.000 | **0.901** | 1.169 | 187.223 | 169.146 | 477.344 | 11.944 | 3.685 | 3.161 | 1.008 |
| MOTIONGRPO$^\aleph$ | **111.776** | **93.702** | **461.702** | **8.330** | **1.309** | 0.963 | **0.399** | **184.992** | **167.032** | **378.423** | **11.886** | **1.614** | **3.156** | **0.199** |

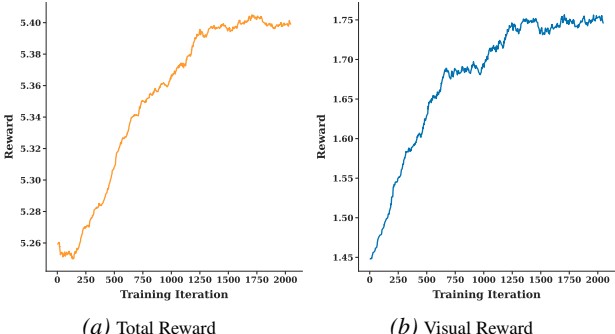

*(a)* Total Reward       *(b)* Visual Reward

*Figure 3.* **Visualization of Reward Curves.** After applying GRPO algorithm, both the total and visual reward values increase.

the MPJPE from 124.985 mm to 114.207 mm and the PA-MPJPE from 103.958 mm to 95.512 mm. This superior performance extends to the RICH dataset, where MOTION-GRPO lowers the MPJPE to 187.223 mm and PA-MPJPE to 169.146 mm. Furthermore, we observe consistent improvements in temporal dynamics and local rotation, evidenced by the reduced MPJVE and MPJRE scores across both datasets (e.g., reaching 531.217 mm/s and 8.413° on AMASS). This reduction in error rates highlights the effectiveness of our approach in resolving kinematic ambiguities and achieving precise joint-level tracking from sparse egocentric signals.

**Visual Plausibility.** Beyond joint alignment, MOTION-GRPO shows superior visual fidelity by effectively mitigating high-frequency artifacts. As detailed in the visual quality metrics, on the AMASS dataset, our method reduces the Jitter score to 2.000, Foot Skating to 1.169 m, and Ground Penetration to 0.901 m compared to EgoAllo. Similarly, in RICH dataset, MOTIONGRPO consistently achieves lower error rates, reducing Jitter from 4.135 to 3.685 and Foot Skating from 1.094 m to 1.008 m. Furthermore, we observe a substantial improvement where our method decreases Ground Penetration from 4.145 m to 3.161 m. These improvements confirm that our joint-level rewards successfully enforce precise tracking, while the trajectory-conditioned perceptual model acts as a robust global filter. This mechanism suppresses unnatural dynamics that standard diffusion constraints often fail to eliminate.

*Table 2.* **Inference Efficiency.** Time and VRAM are per sequence ($T = 128$). $\aleph$ denotes test-time processing.

| Method | Speed(s) | VRAM(GB) |
|---|---|---|
| EgoAllo | 1.150 | 0.65 |
| EgoAllo$^\aleph$ | 1.333 | 1.01 |
| MOTIONGRPO | 1.139 | 0.64 |
| MOTIONGRPO$^\aleph$ | 1.352 | 1.01 |

**Reward Curves.** We further validate our optimization strategy by analyzing the reward curves shown in Fig. 3. Both the total and visual rewards exhibit a consistent upward trend throughout the training iterations. This simultaneous convergence demonstrates that our GRPO-based post-training strategy effectively enhances joint level reconstruction accuracy and visual plausibility of recovered motion.

**Inference Efficiency.** Since MOTIONGRPO utilizes an RL-based post-training strategy to fine-tune the pre-trained diffusion model, the fundamental network architecture remains structurally identical to the base model. The proposed hybrid reward mechanism, including the trajectory-conditioned perceptual model and explicit joint constraints, is employed exclusively during the training phase for advantage estimation and policy updates. Similarly, the temporal noise-injection strategy, introduced to address low intra-group diversity during optimization, is strictly deactivated during the inference stage to ensure deterministic reconstruction. Consequently, our method theoretically imposes zero additional computational overhead or latency compared to the standard diffusion-based reconstruction baselines. The inference process relies solely on the optimized policy network $\pi_\theta$ without requiring access to the reward models or the GRPO value estimation modules.

To empirically validate our efficiency, we report the detailed inference time and memory consumption. As shown in Table 2, MOTIONGRPO maintains comparable inference speed and VRAM usage to the baseline, confirming that our RL-based post-training introduces negligible computational overhead in practical deployment.

*Table 3.* **Ablation Study** on AMASS and RICH datasets. We evaluate Joint Accuracy alongside Visual Quality.

| Method | AMASS Dataset | | | | | | | RICH Dataset | | | | | | |
| | Joint Accuracy | | | | Visual Quality | | | Joint Accuracy | | | | Visual Quality | | |
| | MPJPE (mm)↓ | PA-MPJPE (mm)↓ | MPJVE (mm)↓ | MPJRE (°)↓ | Jitter ↓ | GP (m)↓ | FS (m)↓ | MPJPE (mm)↓ | PA-MPJPE (mm)↓ | MPJVE (mm)↓ | MPJRE (°)↓ | Jitter ↓ | GP (m)↓ | FS (m)↓ |
|---|---|---|---|---|---|---|---|---|---|---|---|---|---|---|
| Baseline | 124.985 | 103.958 | 553.221 | 8.777 | 2.394 | 1.143 | 1.290 | 192.686 | 172.724 | 506.992 | 12.734 | 4.135 | 4.145 | 1.094 |
| + Vanilla GRPO | 117.418 | 97.945 | 543.012 | 8.403 | 2.084 | 0.999 | 1.272 | 190.248 | 171.223 | 494.125 | 12.369 | 3.793 | 3.633 | 1.076 |
| + Visual | 116.549 | 96.729 | 546.128 | 8.427 | 2.014 | 0.886 | 1.221 | 189.103 | 170.002 | 497.480 | 12.268 | 3.684 | 3.225 | 1.044 |
| + Perlin Noise | 114.207 | 95.512 | 531.217 | 8.413 | 2.000 | 0.901 | 1.169 | 187.223 | 169.146 | 477.344 | 11.944 | 3.685 | 3.161 | 1.008 |

*Table 4.* **Ablation Study** on the impact of Perlin noise intensity on intra-group diversity. We perform this evaluation on a subset of the training set (Trumble et al., 2017). "↑" means higher is better.

| Method | Diversity↑ |
|---|---|
| No Noise ($\lambda$=0) | 1.8827 |
| Perlin Noise ($\lambda$=0.05) | 2.2048 |
| Perlin Noise ($\lambda$=0.1) | 3.1430 |

*Table 5.* **Ablation Study** on Accuracy of perceptual model. We report the accuracy (%) and wrong sample count on the AMASS dataset. "$\alpha$" denotes the noise scale. "↓" indicates lower is better.

| Method | Accuracy (%) ↑ | Wrong Count ↓ |
|---|---|---|
| GT Noise ($\alpha$=0.01) | 84.31 | 54 |
| GT Noise ($\alpha$=0.05) | 86.04 | 48 |
| GT Noise ($\alpha$=0.10) | 84.88 | 52 |
| HN-Samples | **97.68** | **8** |

## 4.3. Ablation Studies

**Effectiveness of Vanilla GRPO.** We first validate the efficacy of applying GRPO with joint-level rewards to the motion diffusion backbone. As detailed in Table 3, the introduction of vanilla GRPO yields substantial improvements across all metrics compared to the baseline, notably reducing MPJPE and PA-MPJPE on both the AMASS and RICH benchmarks. Crucially, while these explicit joint rewards primarily focus on local joint precision, the resulting geometric alignment concurrently enhances global motion quality. This result highlights the core advantage of using RL. Unlike standard diffusion training that relies on distribution alignment, GRPO and corresponding joint-level rewards directly optimizes the non-differentiable reconstruction accuracy, providing the fine-grained guidance needed for precise motion recovery.

**Effectiveness of Visual Reward.** We further evaluate the impact of the trajectory-conditioned perceptual model on motion quality. As shown in Table 3, adding the visual reward significantly improves visual quality compared to the "Vanilla GRPO" setting. Specifically, it reduces all Visual Quality metrics across both benchmarks. This improvement is particularly significant on the RICH dataset, which features challenging motions with frequent ground interactions (e.g., push-ups). In such complex scenarios, standard joint-level objectives often fail to provide sufficient global guidance. In contrast, our visual reward explicitly learns to penalize these implausible states. It provides correction where joint-level constraints fall short. These results confirm that our global visual guidance acts as an effective high-level filter. It provides global perceptual quality evaluation to eliminate unrealistic dynamics, ensuring the recovered motion is both accurate and visually natural.

**Effectiveness of Perlin Noise.** We subsequently evaluate

the impact of the Perlin noise-injection strategy used in GRPO. As demonstrated in Table 4, increasing the intensity of Perlin noise effectively amplifies the diversity of generated results, directly mitigating the collapse of group variance. Table 3 further substantiates the impact of this diversity on improving GRPO performance. Injecting temporally smooth noise yields substantial improvements across both joint and visual fidelity metrics. From a joint perspective, the restored gradient flow enhances geometric accuracy, leading to a consistent reduction in pose and rotation errors on both datasets. At the visual level, this stable optimization significantly improves motion quality, effectively mitigating perceptual artifacts. We attribute these gains to the robust gradient signals facilitated by the noise injection. By ensuring a non-trivial standard deviation during group-relative advantage normalization, our strategy allows the policy to achieve finer alignment with the ground truth motion and superior visual fidelity.

**Effectiveness of Hard-Negative Samples.** We evaluate the impact of synthesized hard negative samples on the performance of the perceptual model. We use a similar approach to synthesize negative samples by adding noise to the GT motion parameters (Shen et al., 2025a). As shown in Table 5, simple noise injection yields suboptimal discrimination capabilities even at the best noise scale. In contrast, our method improves accuracy and reduces the instances where generated samples score higher than the GT. These results demonstrate that hard negative samples provide more effective supervision than random perturbations, enabling the model to distinguish between natural and flawed motions.

**Comparison with Other Post-training Paradigms.** To further validate the effectiveness of our proposed framework, we provide experiments comparing the proposed MO-TIONGRPO against other post-training paradigms, specifi-

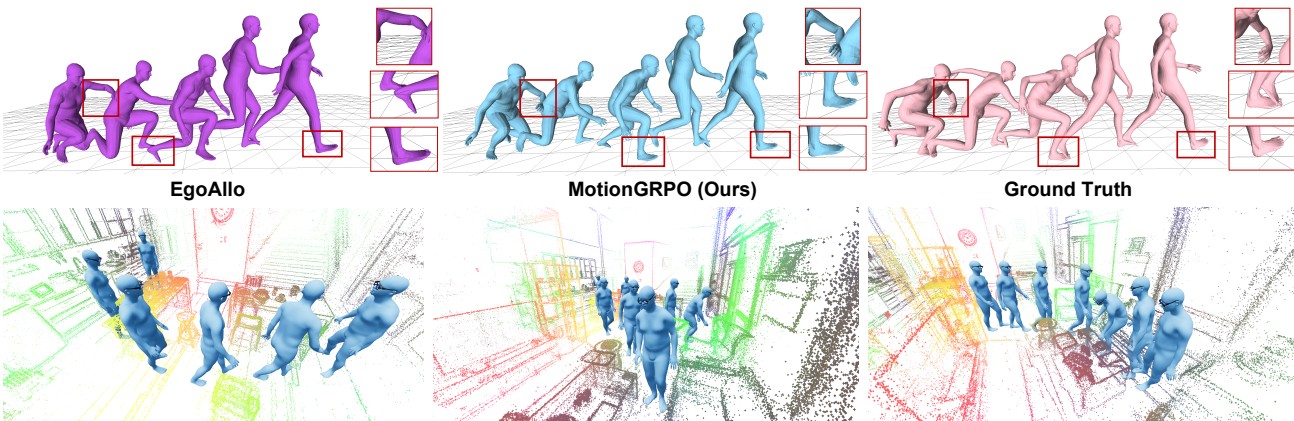

*Figure 4.* **Qualitative Comparison and Visualization.** The first row presents a qualitative comparison with the most competitive baseline on the AMASS dataset. Red boxes highlight failure cases, such as ground penetration and inaccurate joint positions. The second row visualizes the results of MOTIONGRPO on the ADT dataset, demonstrating its generalization capability to real-world scenarios.

*Table 6.* **Quantitative Comparison** with other post-training paradigms using EgoAllo as the baseline on the AMASS dataset.

| Method | Joint Accuracy | | | | Visual Quality | | |
|---|---|---|---|---|---|---|---|
| | MPJPE ↓ | PA-MPJPE ↓ | MPJRE ↓ | MPJVE ↓ | Jitter ↓ | GP ↓ | FS ↓ |
| EgoAllo | 124.985 | 103.958 | 8.733 | 553.221 | 2.394 | 1.143 | 1.290 |
| Fine-Tuning | 124.980 | 103.923 | 9.117 | 543.663 | 2.376 | 1.092 | 1.341 |
| DPO | 118.043 | 100.920 | 8.449 | 538.230 | 2.106 | 1.530 | **1.134** |
| MOTIONGRPO | **114.207** | **95.512** | **8.413** | **531.217** | **2.000** | **0.901** | 1.169 |

cally direct fine-tuning and Direct Preference Optimization (DPO) (Rafailov et al., 2023). Following the same experimental setup, we use EgoAllo as the baseline for all compared methods. The quantitative results on the AMASS dataset are summarized in Table 6.

It is found that direct fine-tuning provides only marginal improvements and even degrades specific visual quality metrics such as Foot Skating. Meanwhile, although DPO yields better results than direct fine-tuning, it still struggles to effectively mitigate spatial and physical artifacts, particularly reflected in local joint reconstruction errors and Ground Penetration. In contrast, our GRPO-based approach achieves the best performance across most of the evaluation metrics. These findings verify that our framework, which optimizes hybrid rewards via GRPO combined with the noise-injection strategy, effectively overcomes the limitations of standard distribution matching. It confirms that our approach is a more robust and effective post-training paradigm for egocentric motion recovery compared to fine-tuning and DPO.

### 4.4. Qualitative Evaluation

Figure 4 presents a qualitative comparison between MOTIONGRPO and the leading baseline. As shown in the first row, our method demonstrates superior robustness under challenging input conditions, effectively mitigating common artifacts such as ground penetration while achieving precise joint reconstruction. The second row visualizes the results of MOTIONGRPO on the ADT dataset, highlighting its capability to faithfully recover human motion

within complex, real-world scene contexts. On ADT, we demonstrate the extensibility of our framework by incorporating visual inputs. Specifically, by integrating a hand estimation model (Pavlakos et al., 2024) consistent with prior work (Yi et al., 2025), we further enhance tracking accuracy and substantiate the scalability of our approach. Additional qualitative results are provided in the Appendix.

## 5. Conclusion

We present MOTIONGRPO, a framework that enhances diffusion-based egocentric motion recovery by integrating RL post-training. By modeling diffusion sampling as a MDP optimized via GRPO, we utilize a hybrid reward mechanism that enforces both global visual plausibility and fine-grained joint precision. Crucially, we identify the "low intra-group diversity" bottleneck inherent in deterministic recovery tasks and introduce a noise-injection strategy to prevent vanishing gradients and stabilize learning. Extensive experiments demonstrate that MOTIONGRPO achieves state-of-the-art performance with superior visual fidelity.

## Acknowledgements

This work is supported by the National Natural Science Foundation of China (No. 62406267), Guangdong Provincial Project (No. 2024QN11X072), Guangzhou-HKUST(GZ) Joint Funding Program (No. 2025A03J3956) and Guangzhou Municipal Education Project (No. 2024312122).

## Impact Statement

This paper presents work whose goal is to advance the field of Machine Learning. There are many potential societal consequences of our work, none of which we feel must be specifically highlighted here.

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

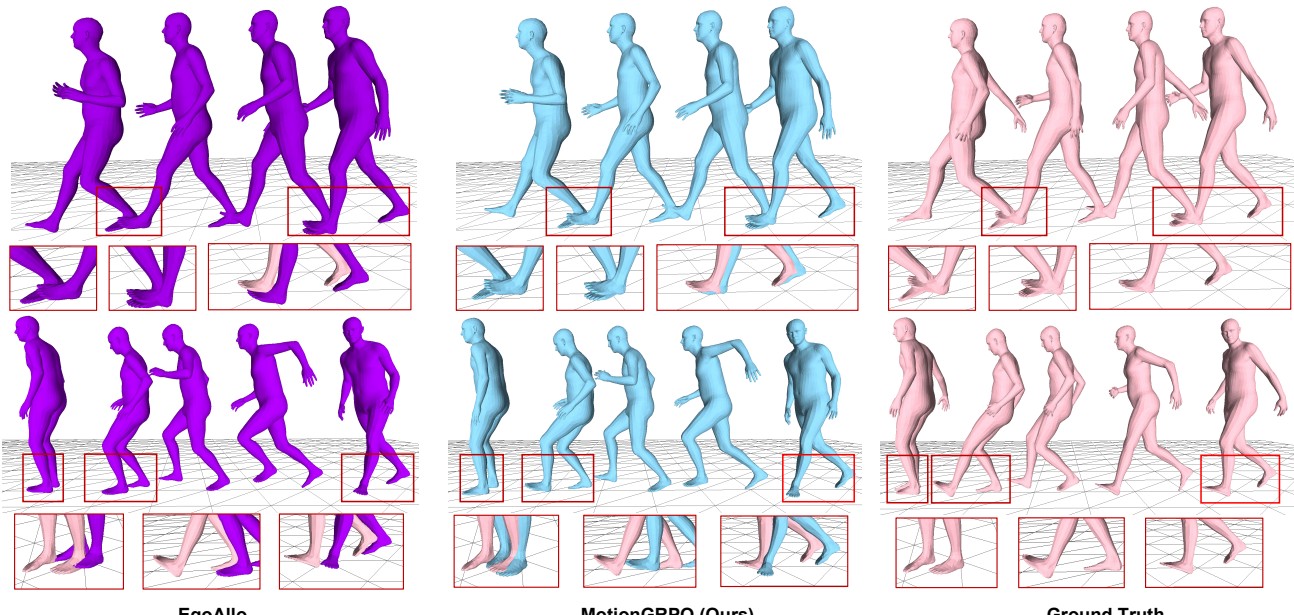

EgoAllo          MotionGRPO (Ours)          Ground Truth

*Figure 5.* **Additional Qualitative Comparisons.** We provide more qualitative comparisons with the most competitive baseline.

## A. Additional Qualitative Comparisons

This section presents supplementary qualitative comparisons against the baseline method, EgoAllo (Yi et al., 2025). As illustrated in Fig. 5, our proposed framework demonstrates superior visual fidelity and achieves tighter alignment with the GT. Notably, while EgoAllo suffers from significant ground penetration and fails to accurately align joints, particularly in highly dynamic sequences with significant global translation, the proposed MOTIONGRPO effectively mitigates these visual implausibilities, resulting in more accurate joint position reconstruction.

## B. Implementation Details

In this section, we provide comprehensive details regarding the implementation of MOTIONGRPO, including the pre-training of the perceptual model, the specific configurations for the GRPO-based post-training, and the settings used during inference.

**Perceptual Model Training.** To train the trajectory-conditioned perceptual model, we utilize a Transformer-based encoder architecture. The input human motion is first processed into an SMPL-H skeleton representation consisting of $N = 21$ body joints, where each joint is represented by a $D = 7$ dimensional feature vector comprising the quaternion rotation and global position. These skeleton features and the corresponding head trajectories are projected into a latent feature dimension of $d = 1024$ using frame-wise and keypoint-wise linear embedding layers. The core network consists of $B = 5$ stacked blocks, where each block sequentially applies MLP, SA and TA layers to capture spatiotemporal dependencies. We employ a cross-attention mechanism to fuse the latent features of the body and head modalities. The model is trained via an online contrastive learning framework (Radford et al., 2021; Li et al., 2022; Ren et al., 2025) using the InfoNCE loss with a temperature parameter $\delta$ set to $0.07$. Particularly, we set **N** to 15, meaning that during training, one set of positive samples and 15 sets of generated negative samples are fed into the model. To ensure robust discrimination capabilities, we synthesize hard negative samples on-the-fly by randomly extracting generated outputs from the last three sampling timesteps of the diffusion process, while GT motion sequences serve as positive samples. The model is optimized using the AdamW (Loshchilov & Hutter, 2019) with a learning rate of $1e - 4$ and a batch size of 16. The training process takes about 8 GPU Hours and 47.2 GB VRAM.

**GRPO Training.** In the post-training phase, we treat the diffusion sampling process as a multi-step MDP to optimize the pre-trained diffusion backbone. We initialize the policy model using the weights of the officially released checkpoint from EgoAllo (Yi et al., 2025). During training, our sequence length remains consistent with that during pre-training, i.e., $T = 128$. To address the issue of low intra-group diversity, we inject temporally smoothed Perlin noise scaled by a factor $\lambda = 0.1$ into the translation components of the input head conditions. For the group of GRPO, we set the group size $G = 16$

and sample outputs via the SDE formulation. The advantage estimation is driven by a hybrid reward mechanism, where the total reward is a weighted sum of the global visual reward derived from the perceptual model and local joint-level rewards, including rotation, position, and velocity constraints. We use exponential normalization for these rewards with specific weight coefficients $\omega_{vis}$, $\omega_{rot}$, $\omega_{pos}$, $\omega_{pos'}$ and $\omega_{vel}$ set to 1.0, 1.0, 1.0, 0.5, and 1.0 respectively. The policy network is updated using a learning rate of $1e-5$ with a batch size of 64. The post-training process takes about 72 GPU hours ($\sim$2000 iteration, about 3 Epoches) and 47.5 GB VRAM.

**Inference.** During inference, the model recovers full-body motion solely from raw head trajectory signals captured by HMDs. For AMASS, we follow the test splits of EgoAllo (Yi et al., 2025). For the RICH dataset, we utilize the standardized test splits, as well as real-world recordings from the ADT dataset. Similarly, we set $T = 128$ as the sequence length during our inference. The generation process begins with a latent representation sampled from a standard Gaussian distribution, which is iteratively denoised over $t = 1000$ timesteps conditioned on the processed head pose invariant features. Unlike the training phase, we do not apply the noise-injection strategy to the head conditions during inference to ensure deterministic and stable reconstruction. The diffusion sampling follows the reverse SDE process, transforming the noisy latents into the final clean motion sequence $\mathbf{M}^{1:T}$. For quantitative evaluation, the generated SMPL-H parameters are converted to mesh representations to compute metrics such as MPJPE and PA-MPJPE, while visual quality is assessed using foot skating and jitter metrics. The quantitative results presented are averaged over 5 runs with different random seeds.

**Hardware.** All experiments are conducted on a server equipped with an AMD EPYC 7542 32-Core Processor and 8 NVIDIA RTX A6000 GPUs (48 GB VRAM).

## C. Dataset Details

In this work, we utilize the AMASS and RICH datasets for training and evaluation. For the AMASS dataset, we strictly adhere to the data splitting strategy employed in EgoAllo (Yi et al., 2025) to ensure a fair and direct comparison with the current state-of-the-art. Specifically, the training set comprises a comprehensive collection of motion capture subsets, including "ACCAD", "BMLhandball", "BMLmovi", "BioMotionLab_NTroje", "CMU", "DFaust_67", "DanceDB", "EKUT", "Eyes_Japan_Dataset", "KIT", "MPI_Limits", "TCD_handMocap", and "TotalCapture". For validation, we utilize "HumanEva", "MPI_HDM05", "MPI_mosh", and "SFU". The test set is strictly reserved to "Transitions_mocap" and "SSM_synced". Regarding the RICH dataset, we note that the authors of EgoAllo did not publicly release the specific details of their evaluation protocol or data splits. To address this and ensure reproducibility, we adopted the official standard test splits provided by the RICH benchmark.

Regarding the ADT dataset, we adhere to the methodology outlined by EgoAllo (Yi et al., 2025). Specifically, for samples containing egocentric imagery, we incorporate a hand pose estimation model (Pavlakos et al., 2024) to serve as a structural prior. These estimates are subsequently utilized during post-processing to refine the final predictions.

## D. Metrics Details

**Local Joint Accuracy.** We report the following metrics to assess joint-level geometric and dynamic precision:

- **Mean Per-Joint Position Error (MPJPE):** This measures the average Euclidean distance between the predicted joint positions $\hat{\mathbf{p}}$ and the ground truth $\mathbf{p}$ across all $N$ joints and $T$ frames:

$$E_{MPJPE} = \frac{1}{T \cdot N} \sum_{\mathcal{T}=1}^{T} \sum_{j=1}^{N} \|\hat{\mathbf{p}}_{\mathcal{T},j} - \mathbf{p}_{\mathcal{T},j}\|_2. \tag{17}$$

- **Procrustes-Aligned MPJPE (PA-MPJPE):** This computes the MPJPE after aligning the predicted pose to the ground truth via Procrustes analysis to factor out global misalignment.

- **Mean Per-Joint Velocity Error (MPJVE):** To assess temporal dynamics, we calculate the deviation in joint velocities. Consistent with the notation, where $v_{\mathcal{T},j}$ denotes the velocity vector, the metric is defined as:

$$E_{MPJVE} = \frac{1}{T \cdot N} \sum_{\mathcal{T}=1}^{T} \sum_{j=1}^{N} \|\hat{\mathbf{v}}_{\mathcal{T},j} - \mathbf{v}_{\mathcal{T},j}\|_2. \tag{18}$$

- **Mean Per-Joint Rotational Error (MPJRE):** This evaluates the orientation accuracy in degrees. While our training objective uses a Euclidean distance on rotation parameters, for standard evaluation, we report the mean geodesic distance between the predicted rotation $\hat{\mathbf{r}}_{t,j}$ and ground truth $\mathbf{r}_{t,j}$:

$$E_{MPJRE} = \frac{1}{T \cdot N} \sum_{\mathcal{T}=1}^{T} \sum_{j=1}^{N} \|\hat{\mathbf{r}}_{\mathcal{T},j} - \mathbf{r}_{\mathcal{T},j}\|_1. \tag{19}$$

**Global Visual Quality.**    To measure visual plausibility, we utilize:

- **Jitter:** Quantifies the smoothness of motion by calculating the average magnitude of the jerk (third derivative of position) for all body joints.

- **Ground Penetration (GP):** Measures physical inconsistency by accumulating the vertical distance of joints penetrating the ground ($z < 0$):

$$E_{GP} = \sum_{\mathcal{T},j} \max(0, -\hat{\mathbf{p}}_z^{(\mathcal{T},j)}). \tag{20}$$

- **Foot Skating (FS):** Evaluates unnatural artifacts by computing the horizontal velocity of foot joints when they are within a ground contact threshold (e.g., height $< 2$ cm).

**Effectiveness of Perceptual Model and Noise.**    To measure the performance of the proposed perceptual model and the noise injection strategy, we utilize:

- **Accuracy & Wrong Count:** These metrics quantify the ability of the trajectory-conditioned perceptual model to distinguish between natural and synthesized motions. Accuracy is formally defined as the proportion of samples where the model assigns a higher score to the GT motion than to the generated counterpart. Let $s_{gt} = \phi(\mathbf{M}_{gt}, \mathbf{c}_{gt})$ and $s_{gen} = \phi(\mathbf{M}_{gen}, \mathbf{c}_{gt})$ represent the plausibility scores given the head condition $\mathbf{c}_{gt}$, where $\phi$ is the perceptual model. The metric is computed as:

$$\text{Accuracy} = \frac{1}{N} \sum_{i=1}^{N} (s_{gt}^{(i)} > s_{gen}^{(i)}). \tag{21}$$

The **Wrong Count** corresponds to the total number of instances where the model fails to identify the GT motion (i.e., $s_{gt} \leq s_{gen}$).

- **Diversity:** We adopt a variant of the diversity metric from MDM (Tevet et al., 2023) to evaluate the intra-group diversity. Distinct from utilizing deep feature embeddings, we compute the average pairwise Euclidean distance directly on the flattened motion representations. This metric measures the variance among the $G$ candidates generated for a single condition:

$$\text{Diversity} = \frac{1}{G(G-1)} \sum_{i=1}^{G} \sum_{j=1,j\neq i}^{G} \|\mathbf{M}_i - \mathbf{M}_j\|_2, \tag{22}$$

where $\mathbf{M}$ represents the flattened vector of the generated motion sequence. A higher diversity score indicates a broader exploration of the solution space, which is essential for effective advantage estimation in GRPO.

# E. Discussion

In this section, we discuss the limitations of our current framework and identify potential directions for future research.

**Environmental Constraints.** MOTIONGRPO currently operates under the assumption of a flat ground plane. This setting strictly follows previous state-of-the-art methods like EgoAllo. Consequently, the model lacks the ability to explicitly perceive or interact with non-planar terrain and scene objects. In real-world scenarios, users frequently interact with furniture or navigate slopes. Our system relies solely on head trajectory signals and body priors. It estimates motion without environmental context. Future work could integrate scene constraints like related work (Guzov et al., 2025) to enable physically consistent interactions with complex environments.

**Training Efficiency.** The training process of GRPO involves a noticeable computational cost. While this algorithm effectively injects guidance, it requires the sampling of a diverse group of outputs at each step. This sampling strategy consumes significant time and computational resources during the training phase. We note that this overhead does not affect the inference latency, which remains comparable to the baseline. However, optimizing the training efficiency remains a critical challenge. Future exploration could investigate more sample-efficient strategies (Shen et al., 2026; Zhang et al., 2026; 2025c) to reduce the resource demands of post-training.

**Broader Applications.** We validate the effectiveness of our framework on a standard transformer-based diffusion architecture. The core principles of our hybrid reward and noise-injection strategy are theoretically model-agnostic. However, we have not yet extended our evaluation to other emerging diffusion-based motion recovery methods or other similar tasks due to limited computational resources. We believe that our approach of leveraging RL to align geometric constraints holds potential for the broader field. Future work will focus on verifying the scalability and performance gains of our method when applied to diverse diffusion backbones.

## F. Downstream Applications

The high-fidelity human motions recovered by MOTIONGRPO offer great potential for various downstream tasks. A primary application is animating digital avatars in VR/AR environments (Shu et al., 2026). Furthermore, our framework can provide a strong pose prior for monocular 3D textured human reconstruction. Recent studies in 3D human reconstruction (Zhang et al., 2025b; Li et al., 2025; Zhang et al., 2025a; Shen et al., 2025b; Zhuang et al., 2025; Yao et al., 2026) aim to build realistic clothed human meshes from images. However, these methods often face challenges due to the ambiguity of vision inputs. This visual ambiguity frequently leads to inaccurate skeleton estimation and incorrect joint positions. Our method accurately recovers human joints and overall body postures from egocentric signals. Therefore, it can offer reliable and stable motion guidance for these reconstruction tasks. By combining our precise motion tracking, future systems can better resolve spatial depth ambiguities. This integration will significantly improve both the geometric accuracy and the visual quality of the reconstructed 3D digital humans.

