# OpenReview forum: "MotionGRPO: Overcoming Low Intra-Group Diversity in GRPO-Based Egocentric Motion Recovery"
_ICML.cc/2026/Conference — ICML 2026 regular_

### Official Review · Reviewer_bCxy · 2026-02-24

**Soundness:** 3
**Presentation:** 3
**Significance:** 2
**Originality:** 2
**Overall Recommendation:** 5
**Confidence:** 1

**Summary:**

The paper presents MOTIONGRPO, a reinforcement learning (RL) post-training framework designed to enhance 3D full-body human motion recovery from head-mounted device (HMD) signals. The authors identify a "low intra-group diversity" bottleneck when applying Group Relative Policy Optimization (GRPO) to motion recovery tasks, where strong conditioning on head trajectories leads to vanishing gradients. To address this, they introduce a temporally smoothed noise-injection strategy using Perlin noise to increase sample variance during training. Additionally, a hybrid reward mechanism is proposed, combining a trajectory-conditioned perceptual model for global visual plausibility with explicit joint-level constraints for local precision. Experimental results on the AMASS and RICH benchmarks demonstrate that the method achieves state-of-the-art performance.

**Compliance With Llm Reviewing Policy:**

Affirmed.

**Final Justification:**

The submission is not in my area.

**Key Questions For Authors:**

1.While the authors emphasize that GRPO avoids the need for a value network, it introduces a significant computational bottleneck due to the online sampling loop, as reflected in the 72 GPU-hour training time. In contrast, Direct Preference Optimization (DPO) is an offline alignment method that avoids iterative sampling during training and typically offers much faster convergence. I recommend the authors provide a comparative study with a DPO-based baseline to justify whether the additional complexity and sampling overhead of MOTIONGRPO yield superior reconstruction accuracy or robustness that cannot be achieved by a more efficient DPO alternative.

2.Regarding the ablation study on negative samples: While using EgoAllo as a hard negative source outperforms random noise, this only addresses the easiness of the task rather than the nature of the learned features. Since a SOTA model like EgoAllo captures many correct physical priors, labeling its entire output as 'negative' may force the perceptual model to overfit to EgoAllo’s specific statistical artifacts rather than learning fundamental physical violations. To clarify this, could the authors provide a 'cross-generator validation'? Specifically, if the perceptual model is used to score samples from a completely different generator architecture (which may lack EgoAllo's specific style artifacts), does it still maintain its discriminative power for physical errors? This is crucial to prove the model is a 'physics-aware perceiver' rather than a 'generator-specific classifier'.

**Limitations:**

Please refer to the "weakness".

**Strengths And Weaknesses:**

**Strength:**

The paper effectively adapts the GRPO algorithm, originally successful in large language model (LLM) reasoning, to the domain of diffusion-based human motion recovery. The hybrid reward mechanism is well-structured, addressing both high-level visual artifacts (like foot skating and jitter) and low-level geometric accuracy (MPJPE). MOTIONGRPO shows statistically significant improvements over competitive baselines like EgoAllo, particularly in reducing MPJPE and PA-MPJPE across multiple datasets. The method maintains a structurally identical network architecture to the base model during inference, imposing zero additional computational overhead or latency.

**Weakness:**

**Comment 1: “GRPO vs. DPO”**

The choice of GRPO over Direct Preference Optimization (DPO) requires further justification. While GRPO eliminates the need for a value network, it necessitates an online sampling process and a "noise-injection" heuristic to function in this domain. It is unclear if a simpler offline alignment method like DPO could achieve similar results without these complexities.

**Comment 2: “The Negative Samples Setting”**

In the perceptual model pre-training, the authors label diffusion-generated samples as "negative." Given that the base model (EgoAllo) is a competitive SOTA, labeling its outputs as fundamentally "negative" suggests a risk of reward hacking, where the policy might learn to avoid specific "style" features of the base model rather than improving physical accuracy.

---

> ### Author Rebuttal · Authors · 2026-03-31
>
> Thanks for recognizing our method's performance and presentation. We address your constructive comments below.
>
> ## W1 & Q1. GRPO vs. DPO and computational cost
>
> ~|MPJPE|PA-MPJPE|MPJRE|MPJVE|Jitter|GP|FS
> -|-|-|-|-|-|-|-
> EgoAllo|124.985|103.958|8.733|553.221|2.394|1.143|1.290
> DPO|118.043|100.920|8.449|538.230|2.106|1.530|1.134
> Ours|**114.207**|**95.512**|**8.413**|**531.217**|**2.000**|**0.901**|**1.169**
>
> We sincerely thank the reviewer for the insightful suggestion to compare our method with DPO and for highlighting the computational bottleneck. Following your insightful suggestion, we implemented a DPO baseline. While DPO is highly efficient: converging in 20 epochs (~4 GPU hours) and significantly faster than our approach. However, our supplementary experimental results show that while DPO improves upon the base model, it falls short of the reconstruction accuracy and visual robustness achieved by MotionGRPO.
>
> We attribute this performance gap to the fundamental difference in how these methods handle reward signals. DPO relies on implicit reward modeling derived from static preference pairs. During this offline process, the model may learn irrelevant features simply to distinguish between good and bad samples. It lacks explicit knowledge of underlying physics. Consequently, DPO struggles to strictly enforce complex kinematic rules.
>
> In contrast, our GRPO framework utilizes an explicit hybrid reward mechanism. We directly compute precise geometric rewards based on joint position, rotation, and velocity, alongside a global visual plausibility score. These explicit metrics provide clear and direct supervision. They actively constrain the generated motion to obey physical rules and align with ground truth states. For a highly constrained task like egocentric motion recovery, this direct geometric guidance is crucial for eliminating local joint errors. Therefore, we believe the significant improvements justify the additional computational overhead during the post-training phase. We will include this comparative study and the corresponding theoretical discussion in the revised manuscript.
>
> ### Implementation Details of DPO
> To construct the static preference dataset for the offline DPO baseline, we treat the GT sequences as the chosen positive samples. The corresponding rejected negative samples are synthesized by explicitly injecting random Gaussian noise into the joint rotations and shape parameters of the GT sequences.
> Let $x^w$ and $x^l$ denote the chosen and rejected sequences respectively. During the training process, we sample a shared diffusion timestep $t$ and Gaussian noise $\epsilon$ to compute the per-sample denoising loss for both the active policy model $\theta$ and the reference model $\theta_{ref}$. Following the standard diffusion preference optimization formulation, the implicit reward for a given sample $x$ is defined as the difference in denoising errors between the reference and policy models, formulated as $r(x)=\mathcal{L}\_{\theta\_{ref}}(x,t,\epsilon)-\mathcal{L}\_{\theta}(x,t,\epsilon)$. We then optimize the policy network using the Bradley-Terry preference framework by minimizing the negative log-sigmoid of the scaled reward margin, expressed as $\mathcal{L}\_{DPO}=-\mathbb{E}\_{(x^w,x^l)}[\log \sigma(\beta(r(x^w)-r(x^l)))]$, where $\beta$ is the hyperparameter controlling the deviation from the reference model.
>
>
> ## W2 & Q2. Cross-generator validation and negative samples
>
> ~|ACC|WC
> -|-|-
> EgoAllo|97.68|8
> Egoego|98.54|5
> EgoAllo'|97.09|10
>
> We appreciate the insightful question regarding whether our perceptual model overfits to generator-specific artifacts rather than learning true physical violations.
> To address this, we evaluate our perceptual model on samples generated by an entirely different architecture, EgoEgo. The evaluation shows that our reward model maintains a highly comparable discrimination accuracy on this unseen architecture, successfully identifying unnatural dynamics errors.
> Furthermore, to reduce the possibility that inherent performance gaps between different baselines might introduce evaluation bias, we conduct an additional rigorous validation. We substantially modify the core architecture of EgoAllo (denoted as EgoAllo' in the table) by altering key structural parameters, such as the number of encoder and decoder layers and the latent space dimensions. After re-training this variant to achieve comparable motion recovery performance, we test our perceptual model on its outputs. The discrimination accuracy remains consistently high and stable without performance degradation. These cross-validation experiments prove that our perceptual model successfully generalizes to capture fundamental visual artifacts, rather than acting merely as a generator-specific scorer.

---

> > ### Author Rebuttal · Reviewer_bCxy · 2026-04-01
> >
> > I will keep my score.

---

### Official Review · Reviewer_AaYj · 2026-03-06

**Soundness:** 2
**Presentation:** 3
**Significance:** 3
**Originality:** 3
**Overall Recommendation:** 4
**Confidence:** 4

**Summary:**

This paper proposes MOTIONGRPO, a post-training framework based on reinforcement learning, for three-dimensional human motion recovery by improving diffusion-based egocentrism. The artifact problem that may occur in the diffusion model is solved by modeling the diffusion sampling process as a Markov decision process. This paper also utilizes Group Relative Strategy Optimization (GRPO) and a hybrid reward mechanism to optimize the credibility of global vision and the accuracy of local joints.

**Compliance With Llm Reviewing Policy:**

Affirmed.

**Final Justification:**

I would like to thank the authors for addressing my questions. While the ablation study on reward coefficients does show the model's robustness, the sensitivity of the MPJPE metric to these rewards appears to be quite low. However, in light of the overall methodological contributions presented in the paper, my final assessment remains positive score.

**Key Questions For Authors:**

1. This paper assumes the occurrence of the vanishing gradient problem. Could some empirical evidence or theoretical proof be provided to support this？
2. To what extent is the final model's performance sensitive to the specific coefficients used in the mixed reward formulation？

**Limitations:**

yes

**Strengths And Weaknesses:**

Strengths
1. This paper proposes a post-training framework based on reinforcement learning to improve diffusion-based motion recovery. Reduce the foot slip, motion jitter, and ground penetration that are common in motion recovery tasks with diffusion models.
2.  Mixed rewards party compensates for the geometric index (joint rotation, position and velocity) and the overall quality gap between movement.


Weaknesses
1. Line 90 of the manuscript describes that directly applying GRPO to diffusion-based motion recovery would encounter the problem of vanishing gradients. However, this paper lacks direct empirical proof of the vanishing gradient problem.
2. This paper is about motion recovery based on head signals. In the case where there is a one-to-one correspondence between head signals and motion, adding Perlin noise can indeed make the generated motion more diverse. However, will this lead to a decrease in the accuracy of the motion recovered from the head signals?
3. This framework assumes a flat ground plane for the experiments, which limits its applicability in in-the-wild scenarios. It remains unclear whether the reward model can still perform well in more challenging motion recovery tasks, such as climbing stairs or sitting on a chair.
4. The comparison method in the experimental part of this article merely compares EgoAllo and EgoEgo, and does not compare with some popular reward-guided methods.

---

> ### Author Rebuttal · Authors · 2026-03-31
>
> Thanks for your valuable feedback and constructive suggestions regarding our manuscript. We deeply appreciate your careful review and address your comments and questions below.
>
> ## W1 & Q1. Empirical proof of the vanishing gradient
> The problem of gradient vanishing is indirectly reflected in Tables 2 and 3 of the manuscript. Specifically, Table 3 proves that without noise-injection strategy, the intra-group diversity remains distinctly low. As detailed in methodology, such limited variance causes the rewards for generated motions within a single group to be nearly identical. This uniformity pushes the standard deviation term in the advantage numerator toward zero, which creates numerical instability and leads to the vanishing gradient problem. Consequently, the optimization objective fails to provide meaningful policy gradient updates.
>
> The empirical evidence of gradient vanishing is shown in Table 2, where applying "vanilla GRPO+visual" yields constrained performance improvements over the baseline. By introducing noise to simulate pseudo OOD inputs, we increase uncertainty and output diversity. This variance ensures a non-trivial standard deviation during advantage calculation. Due to this stabilized gradient flow, our full model with noise significantly outperforms the vanilla setup across all metrics.
>
> ## W2. Impact of Perlin noise on motion accuracy
> We clarify that the noise is exclusively applied to the head trajectory conditions during the training phase to simulate pseudo OOD inputs. This procedure is essential to restore output variance and mitigate the vanishing gradient problem induced by low intra-group diversity. Therefore, this strategy does not reduce the model's accuracy, but improves the model's performance.
>
> Table 2 proves that noise enhances overall reconstruction accuracy. By maintaining a non-trivial standard deviation for advantage normalization, the robust gradient flow allows the policy network to achieve finer geometric alignment with the GT, leading to a reduction in errors across our evaluated datasets without compromising the one-to-one correspondence during deployment.
>
> ## W3. Applicability to non-flat scenarios
> We acknowledge that our framework currently operates under the assumption of a flat ground plane. This setting follows EgoAllo to ensure fair and direct comparisons. However, our proposed hybrid reward mechanism is theoretically adaptable to complex environments.
>
> Given a complex 3D scene and a motion sequence, our perceptual model could be augmented with a 3D scene backbone, such as PointNet, to extract scene features and integrate them via cross-attention. During the training of the perceptual model, motions that exhibit severe scene penetration would be utilized as negative samples. This augmented reward model would penalize physically implausible sequences during optimization, guiding the network toward valid human-scene interactions without altering the core framework. We take this task as our future work.
>
> ## W4. Comparison with other reward-guided methods
> To our best knowledge, MotionGRPO is the **first** framework to introduce a reward-guided method specifically to the task of egocentric human motion recovery from sparse HMD signals.
> While reward-guided methods are popular in other domains, e.g., unconstrained motion generation, directly transferring them to this task presents fundamental challenges. As detailed in our methodology, egocentric motion recovery is heavily constrained by head trajectory inputs. This conditioning leads to low intra-group sample diversity and vanishing gradients during standard policy optimization, making popular reward mechanisms ineffective.
> Furthermore, due to the required substantial engineering, we will include this comparison in the final version.
>
> ## Q2. Sensitivity to reward coefficients
>
> ~|MPJPE|PA-MPJPE|MPJRE|MPJVE|Jitter|GP|FS
> -|-|-|-|-|-|-|-
> Ours|114.207|95.512|8.413|531.217|2.000|0.901|1.169
> $R_{jrt}$=0.5|114.623|95.703|8.501|533.217|1.999|0.872|1.154
> $R_{vis}$=0.5|113.555|95.381|8.412|530.524|2.014|0.918|1.188
>
> Different from the original setting, where $R_{jrt} = R_{vis} =1.0$, we conduct additional ablation studies varying the relative weights of the local joint reward and the global visual reward. As shown in the Table above, the experimental results prove that the final model's performance is generally robust to coefficient changes, remaining stable without great degradation across all metrics. The specific weights effectively control the trade-off between strict geometric alignment and visual fidelity. Specifically, reducing the weight of the joint-level reward leads to a slight decrease in pose and rotation accuracy while maintaining strong visual quality. Conversely, reducing the visual reward coefficient marginally improves strict joint tracking but results in slight decrease across visual metrics.

---

> > ### Author Rebuttal · Reviewer_AaYj · 2026-04-01
> >
> > I would like to thank the authors for addressing my questions. While the ablation study on reward coefficients does show the model's robustness, the sensitivity of the MPJPE metric to these rewards appears to be quite low. However, in light of the overall methodological contributions presented in the paper, my final assessment remains positive score.

---

### Official Review · Reviewer_ZzkV · 2026-03-09

**Soundness:** 3
**Presentation:** 2
**Significance:** 3
**Originality:** 3
**Overall Recommendation:** 4
**Confidence:** 3

**Summary:**

This paper proposes MOTIONGRPO, a reinforcement learning-based post-training framework for egocentric human motion recovery. The authors formulate the sampling process of a diffusion model as a Markov Decision Process and optimize it with GRPO, while designing a hybrid reward that combines global visual plausibility and local joint accuracy. In addition, the paper points out that GRPO suffers from degraded advantage estimation and vanishing gradients due to low intra-group sample diversity in this task. To address this issue, the authors further introduce temporally smooth noise injection into the input head trajectories to increase intra-group diversity and stabilize training. Experimental results show that the proposed method achieves better joint reconstruction accuracy and visual quality on the AMASS and RICH datasets.

**Compliance With Llm Reviewing Policy:**

Affirmed.

**Final Justification:**

My concerns are addressed, the Overall Recommendation can be adjusted to weak  accept.

**Key Questions For Authors:**

See  weaknesses  part.

**Limitations:**

Yes

**Strengths And Weaknesses:**

Strengths：
1.	The paper formulates the diffusion sampling process as a Markov Decision Process and introduces GRPO for post-training, leveraging reinforcement learning to improve the quality of the generated results.
2.	By injecting temporally smooth noise to increase intra-group diversity, the method is relatively simple to implement yet effective in improving training stability.
3.	It achieves better performance than the baselines on both AMASS and RICH, while introducing almost no additional overhead during inference.

Weaknesses:

1.$R_{pos}$ and $R^'_{pos}$   are both essentially designed to measure joint position accuracy, which may introduce redundant reward signals. The authors set $w_{pos}$ and $w^'_{pos}$ to 1.0 and 0.5, respectively; is this also intended to balance such redundancy? Please provide additional ablation studies to clarify the individual contributions of these two reward terms.

2.	The noise injection is applied to the conditioning signal rather than introducing diversity in the sampling process itself. If each sample is generated under a different perturbed condition, then it is unclear whether the reward differences across samples reflect the quality of the policy itself or simply the differences in the input conditions.

3.	In Section 3.4.1, the perceptual model is trained using negative samples generated by the base policy model, and is then used as a reward function to further optimize the diffusion model. How do the authors avoid training instability or reward hacking in this closed-loop setup?

4.	The joint-level rewards are closely aligned with the evaluation metrics MPJPE, PA-MPJPE, MPJVE, and MPJRE, which raises the risk of overfitting to the benchmark metrics. The paper also seems to lack ablation studies specifically analyzing the effect of the joint-level rewards.

5.	Since the overall paradigm is diffusion + GRPO, the authors should at least compare against other post-training paradigms to justify the choice of GRPO, such as direct fine-tuning or DPO.

6.	There are many writing and notation issues. In Algorithm 1, Step 11 mixes the index k, using it both as a summation index and as a variable. A similar notation issue appears with the symbol T in Section 3.4.1. There are also clear typos or formatting problems, such as the symbol “⊔” in Eq. (13), “focus” in line 79, “an Stochastic” in line 75, and “a online” in line 218.

---

> ### Author Rebuttal · Authors · 2026-03-31
>
> Thanks for your careful review and feedback. We appreciate your insightful comments and address them below.
>
> ## Q1. Ablation of Rewards
> The weights were specifically chosen to balance reward signals and prevent the model from over-prioritizing joint position.
> While both terms evaluate positional accuracy, they offer complementary reward signals. $R_{pos}$ computes the error without alignment to focus on spatial positioning. Conversely, $R'_{pos}$ applies alignment to focus on fine-grained, joint pose accuracy independent of spatial drift.
>
> ~|MPJPE|PAMPJPE
> -|-|-
> w/o $R_{pos}$|116.085|97.388
> w/o $R'_{pos}$|115.896|97.484
>
> The ablation studies are detailed above. The specific metric drops align with the design of the two rewards. Removing $R_{pos}$ causes a greater degradation in MPJPE, confirming its role in spatial positioning. Conversely, removing $R'_{pos}$ has a greater negative impact on PA-MPJPE, verifying its contribution to fine-grained pose alignment. Therefore, these two terms focus on different aspects of pose accuracy and are mutually beneficial rather than redundant.
>
> ## Q2. Noise injection on conditioning signals
> We clarify that we do not use different perturbed conditions for samples within the same group. Following standard GRPO, all samples in a single group share the exact same perturbed condition. We apply noise to the raw head trajectory only before passing it to the conditioning function. Therefore, the reward differences across samples strictly reflect the quality of the policy itself, rather than differences in conditions. The necessary intra-group diversity comes entirely from the stochasticity of the SDE sampling. This design ensures stable advantage estimation. Our ablations (Table 2, w/ & w/o noise) confirm that the strategy restores gradient flow and improves overall performance.
>
>
> ## Q3. Preventing reward hacking and instability
> We clarify that the setting is not a fully closed-loop system. For the perceptual model's training, we sample negative samples from the *last three timesteps* of the base policy. The GRPO optimization strictly calculates rewards using the *final denoised output (t=0)*. This explicit mismatch prevents the diffusion model from exploiting the reward model, ensuring stable optimization.
>
> Regarding training instability, our GRPO formulation uses standard techniques, including KL divergence penalties and clipping terms, to strictly constrain the update steps. The training curve in Fig.3 proves the stability of the training process.
>
> Regarding reward hacking, as demonstrated in previous research [1], combining multiple rewards can alleviate reward hacking. By jointly optimizing the hybrid rewards, we force the policy to balance multiple objectives, making it difficult to exploit a single reward component without incurring penalties from the others. Additionally, tricks such as assigning shared initialization noise within the group during the sampling, as introduced in DanceGRPO, can be easily integrated to suppress reward hacking.
>
> [1] Amodei et al., Concrete Problems in AI Safety
>
> ## Q4. Risk of overfitting and joint-level reward ablation
> Regarding the ablation study, we clarify that it has already been presented in Table 2 of our manuscript. Specifically, the setting denoted as "+Vanilla GRPO" serves as the ablation study for the joint-level rewards, as it applies only the joint reward to the backbone during training. The comparison between the "Baseline" and "+Vanilla GRPO" settings proves the effectiveness of the proposed joint-level guidance.
>
> Furthermore, addressing the concern of metric overfitting, "Baseline" and "+Vanilla GRPO" comparison illustrates that optimizing the joint-level rewards not only improves the corresponding metrics but also enhances the visual quality. This indicates the geometric alignment provided by the joint constraints yields visually plausible motions, rather than merely exploiting the mathematical formulations of the metrics. Additionally, generalization on ADT proves the framework learns transferable priors instead of just overfitting.
>
> ## Q5. Comparison with other post-training paradigms
> We provide experiments comparing our method against direct FT and DPO:
> ~|MPJPE|PA-MPJPE|MPJRE|MPJVE|Jitter|GP|FS
> -|-|-|-|-|-|-|-
> FT|124.980|103.923|9.117|543.663|2.376|1.092|1.341
> DPO|118.043|100.92|8.449|538.230|2.106|1.530|**1.134**
> Ours|**114.207**|**95.512**|**8.413**|**531.217**|**2.000**|**0.901**|1.169
>
> As shown above, direct FT provides marginal improvements and even degrades specific metrics (FS). DPO struggles to effectively mitigate artifacts (joint error & GP). Our GRPO-based approach achieves the best performance across most of the metrics. These findings confirm that optimizing hybrid rewards via GRPO is the most effective paradigm.
> (Please refer to Q1 of Reviewer bCxy to see the implementation of DPO)
>
> ## Q6. Writing & typos
> We sincerely apologize for these errors. We will ensure all notations are consistent in the final version

---

> > ### Author Rebuttal · Reviewer_ZzkV · 2026-04-05
> >
> > The authors have solved my concerns,  the score can be adjusted.

---

### Official Review · Reviewer_Sg3Z · 2026-03-12

**Soundness:** 3
**Presentation:** 3
**Significance:** 3
**Originality:** 3
**Overall Recommendation:** 5
**Confidence:** 4

**Summary:**

The authors propose a method to predict 3D human pose and motion from the trajectory of the head, e.g., tracked with an IMU or vision-based by SLAM. The basis is a diffusion policy RL method that build upon the GRPO method popular for LLM training. Extensions are proposed to make it applicable to this setting, by enhancing diversity of samples with temporally stable Perlin noise. The authors also propose a "Visual Level Reward" that scores/embeds the realism of a motion based on pre-training with contrastive learning on sequences from the base policy.

**Compliance With Llm Reviewing Policy:**

Affirmed.

**Final Justification:**

Thanks, this fully addresses my concerns. I'll raise my score to accept.

**Key Questions For Authors:**

* how many epochs/iteration is the method trained. Is there some form of early stopping?
* please comment on the visual vs. geometric discussion

**Limitations:**

Limitations are given in detail in the supplemental but mentioning some would be good in the main paper.

**Strengths And Weaknesses:**

Significance:
* It addresses an important problem as augmented reality devices are becoming more popular but equipping them with cameras provides privacy concerns. Just predicting motion from IMU-based head motion is therefor very relevant. The deployed techniques are also very timely and contribution to knowlede in the applied ML domain. The method is not particularly original but it does combine existing building blocks creatively and patches them up where necessary.

Weaknesses:
* The method is generally sound, yet I found the labeling of the visual reward as misleading. When reading it and looking at the teaser, I thought it would be linked to images or renderings. It is more a 3D gemetry aware encoding and I would recommend renaming it as such.
* 3.4.2. JOINT LEVEL REWARD is a bit scarce on information. It would be nice to link to the supplemental.
* the related work or introduction could better distinguish from egocentric methods using cameras, e.g., [1] and follow up work. It would even be interesting to evaluate on their datasets to see the remaining gap to image-conditioned and head-conditioned models.
* missed at least some basic implementation details in the main paper. Do I need a GPU cluster, will it take weeks or hours to train? Details are given in the supplemental in sufficient depth.

Strengths:
* The approach combines very recent advances, including diffusion and specific RL models with classical Perlin noise, creating a novel approach for prediction. I wondered a bit how general these advances are and whether they would apply more broadly.
* The method is evaluated on established benchmarks and chooses a fair environment to compare to prior work. Metrics are extensive, including foot skating and penetration.
* The  main contributions are ablated and show clear improvements
[1] Rhodin, H., Richardt, C., Casas, D., Insafutdinov, E., Shafiei, M., Seidel, H.P., Schiele, B. and Theobalt, C., 2016. Egocap: egocentric marker-less motion capture with two fisheye cameras. ACM Transactions on Graphics (TOG), 35(6), pp.1-11.

---

> ### Author Rebuttal · Authors · 2026-03-31
>
> Thanks for recognizing the soundness of our method. We deeply appreciate your constructive comments and address your questions and suggestions below.
>
> ## W1 & Q2. Clarification on "Visual Reward" vs. Geometric Encoding
> We appreciate the reviewer's constructive feedback regarding the terminology of the "Visual Level Reward." We agree that the current label may lead to expectations of image-based rendering. In the revised manuscript, we will clarify this distinction and consider more precise terminology to better reflect its role in ensuring global motion fidelity.
>
> While this reward is computed from 3D skeleton sequences rather than 2D rendering or pixels, we designated it as "visual" because its primary objective is to evaluate the global visual plausibility and identify artifacts that are most salient to a human observer, such as foot skating, motion jitter, and ground penetration. Our experiments demonstrate that while explicit joint-level rewards prioritize local precision, the trajectory-conditioned perceptual model effectively serves as a high-level filter to mitigate these visually implausible dynamics that standard geometric losses often fail to suppress.
>
> ## W2. Details on Joint Level Reward
> We appreciate the feedback. We will expand the description of the Joint Level Reward in Section 3.4.2 of the main paper to provide better context, and we will explicitly add a pointer to the Appendix for the comprehensive mathematical formulation and details.
>
> ## W3. Distinction from Camera-based Methods
> We sincerely thank the reviewer for the constructive suggestion. We will expand the Introduction and Related Work sections to explicitly distinguish our head-conditioned approach from camera-based egocentric methods in the final version.
>
> Principally, camera-based methods leverage direct visual evidence of the user's body and surroundings, which inherently provides strong geometric constraints for pose estimation. In contrast, our approach operates in a severely under-constrained setting, recovering full-body motion solely from sparse head trajectory signals, which requires resolving significant kinematic ambiguities primarily through generative priors. This distinction in input modality leads to our specific technical focus: addressing the lack of fine-grained local control in diffusion priors by injecting explicit geometric and visual guidance via RL. Regarding the evaluation gap, as suggested, we will evaluate our method on the vision datasets such as Egocap, Nymeria and GIMO datasets utilized by related works. This added comparison and discussion will demonstrate the remaining performance gap between image-conditioned and head-only models, providing a clearer perspective on the boundaries of our proposed framework and the inherent challenges of sparse-signal recovery.
>
> ## W4 & Q1. Training and implementation details
> We thank the reviewer for the constructive feedback regarding the implementation details. We will ensure that a summary of the hardware requirements and training duration is moved from the supplementary material into the main paper for the final version.
>
> To address your specific questions, our method does not strictly require a massive GPU cluster, but it does require high-memory GPUs. As mentioned in the Appendix, all the experiments are conducted in a server with 8 A6000 NVIDIA GPUs. Regarding the training duration and stopping criteria, the model is optimized until the reward values converge, which generally occurs at approximately 2000 iterations (3 Epoches) as illustrated in our reward curve visualizations. We do not employ a traditional early stopping mechanism monitored via a separate validation loss; instead, the training naturally stops when the rewards stabilize after demonstrating a consistent upward trend.

---

> > ### Author Rebuttal · Reviewer_Sg3Z · 2026-04-03
> >
> > Thanks, this fully addresses my concerns. I'll raise my score to accept.

---

### Decision · Program_Chairs · 2026-04-30

**Decision:**

Accept (regular)

**Comment:**

This paper proposes MotionGRPO, a reinforcement learning post-training framework for diffusion-based egocentric 3D human motion recovery from head-mounted device signals. The key contributions include: (1) formulating diffusion sampling as a Markov Decision Process optimized with GRPO; (2) a hybrid reward combining a trajectory-conditioned perceptual model for global visual plausibility and explicit joint-level constraints; and (3) a Perlin noise injection strategy to address the vanishing gradient problem caused by low intra-group sample diversity in this highly conditioned setting. The method achieves state-of-the-art results on AMASS and RICH benchmarks with no additional inference overhead.

The paper received four reviews: two Accept and two Weak Accept. After the rebuttal, all four reviewers acknowledged their concerns as fully resolved.

Overall, the paper makes a solid technical contribution by identifying and addressing a specific and non-obvious challenge (low intra-group diversity) when applying GRPO to a highly conditioned diffusion task. The proposed solution is well-motivated, the experimental evaluation is comprehensive, and the rebuttal successfully addressed the substantive concerns raised by all reviewers. The post-rebuttal consensus is uniformly positive. The work is likely to be of interest to both the egocentric motion recovery and the RL-for-generative-models communities.

Therefore, the AC is happy to accpet this paper to ICML. Congratulations! Please well incorporate the reviewers' feedback to prepare a solid camera-ready version.